A probabilistic analysis reveals fundamental limitations with the environmental impact quotient and similar systems for rating pesticide risks

Peterson Robert K.D. bpeterson@montana.edu
Schleier Jerome J. III
Department of Land Resources and Environmental Sciences, Montana State University , Bozeman, Montana , USA
Backhaus Thomas
Electronic publication date: 2014 Apr 22
Publication date: 2014
Volume: 2
Electronic Location ID: e364
Received 2014 Jan 15; Accepted 2014 Apr 6
Copyright: © 2014 Peterson and Schleier III
Copyright year: 2014
Copyright holder: Peterson and Schleier III
License: This is an open access article distributed under the terms of the Creative Commons Attribution License, which permits unrestricted use, distribution, and reproduction in any medium, provided the original author and source are credited.
License URL: https://creativecommons.org/licenses/by/3.0/

Keywords: Risk ranking, Integrated pest management, Comparative risk assessment, Exposure assessment, Risk analysis, Pesticide

Funding: Montana Agricultural Experiment Station and Montana State University This study was supported in part by the Montana Agricultural Experiment Station and Montana State University. The funders had no role in study design, data collection and analysis, decision to publish, or preparation of the manuscript.

==============================
Comparing risks among pesticides has substantial utility for decision makers. However, if rating schemes to compare risks are to be used, they must be conceptually and mathematically sound. We address limitations with pesticide risk rating schemes by examining in particular the Environmental Impact Quotient (EIQ) using, for the first time, a probabilistic analytic technique. To demonstrate the consequences of mapping discrete risk ratings to probabilities, adjusted EIQs were calculated for a group of 20 insecticides in four chemical classes. Using Monte Carlo simulation, adjusted EIQs were determined under different hypothetical scenarios by incorporating probability ranges. The analysis revealed that pesticides that have different EIQs, and therefore different putative environmental effects, actually may be no different when incorporating uncertainty. The EIQ equation cannot take into account uncertainty the way that it is structured and provide reliable quotients of pesticide impact. The EIQ also is inconsistent with the accepted notion of risk as a joint probability of toxicity and exposure. Therefore, our results suggest that the EIQ and other similar schemes be discontinued in favor of conceptually sound schemes to estimate risk that rely on proper integration of toxicity and exposure information.

Introduction

Numerous methods to rate pesticide risks have been introduced over the past two decades. The methods are typically qualitative or semi-quantitative and involve rating and weighting hazard, toxicity, and exposure factors for pesticide active ingredients. The purpose of these rating schemes is to provide growers and other decision makers with information so that they can discriminate among pesticides based on their risk to such entities as people, other non-target organisms, and water quality.

Comparing risks among pesticides has substantial utility for decision makers (Peterson, 2006). These comparisons are needed in addition to risk assessments of specific pesticides by regulatory agencies. A regulatory agency, such as the U.S. Environmental Protection Agency, should not be the sole arbiter of risk information and management decisions about pesticides. However, if rating schemes to compare risks from pesticides are to be used, they must be conceptually and mathematically sound.

The most influential scheme is arguably the Environmental Impact Quotient (EIQ) by Kovach et al. (1992). Since the introduction of the EIQ, numerous researchers have evaluated it or adapted it for their own risk rating schemes, or both (Cross & Edwards-Jones, 2011; Finizio, Calliera & Vighi, 2001; Greitens & Day, 2007; Higley & Wintersteen, 1992; Labite, Butler & Cummins, 2011; Leach & Mumford, 2011; Maud, Edwards-Jones & Quin, 2001; Muhammetoglu, Durmaz & Uslu, 2010; Muhammetoglu & Uslu, 2007; Reus et al., 2002; Reus & Leendertse, 2000; Sande et al., 2011; Stenrod et al., 2008; Surgan, Condon & Cox, 2010; van der Werf, 1996; Vercruysse & Steurbaut, 2002; Yazgan & Tanik, 2005). In addition, EIQs for pesticides continue to be updated on a dedicated web site of the New York State Integrated Pest Management Program, Cornell University (www.nysipm.cornell.edu/publications/eiq/).

The EIQ method essentially is a mathematical formula that determines environmental impact for pesticide active ingredients based on converting a raft of physicochemical and toxicological information, such as acute dermal toxicity, toxicity to birds, long-term health effects, and soil runoff potential, into an arbitrary ratings scale of 1, 3, and 5 and then combining and weighting those ratings through multiplication, division, and addition. This computation results in EIQs for farm worker, consumer, and environment. The EIQs from these three component categories are then averaged to determine a total EIQ. The EIQ equation is: EIQ=CDT∗5+DT∗P+C∗S+P/2∗SY+L+ F∗R+DS+P/2∗3+Z∗P∗3+B∗P∗5/3

where: C, chronic toxicity; DT, dermal toxicity; P, plant surface half-life; S, soil half-life; SY, systemicity; L, leaching potential; F, fish toxicity; R, surface loss potential; D, bird toxicity; Z, bee toxicity; B, beneficial arthropod toxicity.

Dushoff, Caldwell & Mohler (1994) critiqued the EIQ method, pointing out several conceptual problems with the approach. Some shortcomings in the method were addressed in the original publication (Kovach et al., 1992) and the problems discussed by Dushoff, Caldwell & Mohler (1994) were recognized by Levitan, Merwin & Kovach (1995). The critique by Dushoff, Caldwell & Mohler (1994) is compelling and suggests that the EIQ method is substantially limited solely on the basis of conceptual problems with scaling and weighting of the rating factors.

Cox, Babayev & Huber (2005) demonstrated mathematically that qualitative risk rating systems are fundamentally limited because they do not adequately incorporate the key risk concept of uncertainty. There are two major problems with qualitative risk rating systems: reversed rankings and uninformative ratings. Reversed rankings occur when assigning a higher qualitative risk rating to situations that have a lower quantitative risk. Uninformative ratings occur when assigning the same qualitative ratings to risks that differ by many orders of magnitude. These major limitations often obscure risk comparisons such that they are unable to distinguish between risks. Moreover, Cox, Babayev & Huber (2005) argue that no consistent quantitative interpretation of qualitative labels is possible and no change in how attributes are rated qualitatively can ensure that a qualitative rating system will give accurate results (but see Levine (2012) for a potential solution using logarithmic scaling). Cox, Babayev & Huber (2005) argue that because of this, quantitative risk models should be used instead of qualitative risk models. Since 2005, Cox and others have expanded the analysis of risk rating systems (Barends et al., 2012; Cox, 2008a; Cox, 2008b; Cox, 2009a; Cox, 2009b; Levine, 2012; Schleier III & Peterson, 2010; Schleier III, Sing & Peterson, 2008).

Here, we examine pesticide risk rating schemes and the EIQ in particular using, for the first time, a probabilistic analytic technique. Our purpose is not to repeat the mathematical proofs of Cox, Babayev & Huber (2005) that clearly demonstrate, sensu lato, fundamental problems of qualitative risk rating schemes. Rather, we will discuss how the problems extend to the EIQ using an approach different from that taken by Dushoff, Caldwell & Mohler (1994). Furthermore, we discuss the discontinuation of the EIQ and other similar schemes in favor of conceptually sound schemes to estimate risk that rely on proper integration of toxicity and exposure information.

Methods

The ratings of 1, 3, and 5 in the EIQ method are surrogates for low, medium, and high risk or impact or toxicity or persistence, depending on the factor of interest. For demonstration purposes only, we show how converting the ratings to estimates of risk probabilities for only four of the factors limits the value of the EIQ method. The EIQ factors, “long-term health effects”, “leaching potential”, and “surface runoff potential”, and ratings of “little-none”, “possible”, “definite”, “small”, “medium”, and “large” imply that they are risks. Therefore, they have a probability of occurrence rather than an absolute certainty of occurring. Similarly, the factor “beneficial arthropod toxicity” has ratings of “low impact”, “moderate impact”, and “severe impact”. Degrees of impact also have associated uncertainty.

Because the ratings of 1, 3, and 5 are surrogates for risk, they can be converted to risk intervals that incorporate the underlying probabilities. Therefore, the simplest, yet coarse, way to do this is to assume the ratings of 1, 3, and 5 span the range of risk from 0 to 1 (or 0 to 100%). A rating of 1, when mapped onto an interval of risks would be 0 to 0.32. A score of 3 would be 0.33 to 0.66 and a score of 5 would be 0.67 to 1. Consequently, if a pesticide has a “surface runoff potential” factor that has a score of 3, it is at medium risk of runoff. However, a discrete score of 3 does not capture the probabilistic nature of risk, yet the score of 3 is intended to represent medium risk. Therefore, the score needs to be mapped to an estimate of risk. This can be done most simply by assuming a uniform probability density function of risk values from 0.32 to 0.66 for medium risk. Medium risk implies uncertainty and probability, but a score of 3 does not accommodate that risk estimate. An interval of 0.33 to 0.66, however crudely, accommodates the probability of occurrence.

To demonstrate the consequences of mapping discrete risk ratings to probabilities, we calculated adjusted EIQs for a group of 20 actual insecticide active ingredients with unadjusted EIQs ranging from 22.1 (methiocarb) to 44 (diazinon). The insecticides evaluated were chosen randomly from lists of active ingredients in Yu (2008), who provides a relatively complete list of currently registered insecticides. Five insecticides each were chosen randomly from four chemical classes: carbamates, neonicotinoids, organophosphates, and pyrethroids. The unadjusted EIQs and ratings were obtained from the New York State Integrated Pest Management Program, Cornell University (www.nysipm.cornell.edu/publications/eiq/). The four factors discussed above were converted to probability ranges of risk and all other factors were held constant at their respective deterministic scores. To align those deterministic scores with the probability ranges mapped for the four factors, the ratings were converted to static probabilities proportional to the value of the scores. For example, a score of 3 for fish toxicity was converted to 0.5.

Using Monte Carlo simulation (Oracle Crystal Ball® 11.2, Denver, CO), we calculated adjusted EIQs under different hypothetical scenarios by incorporating the probability ranges associated with the four factors (Fig. 1). Probabilities of occurrence of adjusted EIQ values were determined by incorporating sampling from the statistical probability density function of each input variable used to calculate the EIQ. Each of the four input variables was sampled 20,000 times. Then, the variability for each input was propagated into the output of the model so that the output reflected the probability of values that could occur.

Figure 1 Adjusted Environmental Impact Quotient (EIQ) values for 20 insecticides based on probabilistic simulation analysis.

For each bar, the bottom line is the 10th, the middle line is the 50th, and the top line is the 90th percentile value from the simulation. The number at the top of each bar is the original EIQ value. The original EIQ value reported for naled, 49, is incorrect. The correct value is 41.

Results and Discussion

Results demonstrate overlaps of adjusted EIQs for insecticides that have discrete EIQs (Fig. 1). For example, when incorporating uncertainty, adjusted EIQs range from 0.75 to 1.17 for cypermethrin and from 0.68 to 1.05 for acetamiprid. Therefore, more than 90% of the adjusted EIQ values for these two insecticides overlap with each other. Yet, the unadjusted EIQs are 36.4 and 28.7, respectively, a 7.7 EIQ unit difference.

Another example can be shown with imidacloprid and dinotefuran, two neonicotinoid insecticides. The adjusted EIQs range from 0.88 to 1.29 for imidacloprid and 0.65 to 1.04 for dinotefuran. More than 26% of the adjusted EIQ values overlap with each other. The unadjusted EIQs are 36.7 and 22.3, respectively, a 14.4 EIQ unit difference. Consequently, these examples show that pesticides with different EIQs, and therefore different putative environmental effects, actually may not be different because of the potential overlap in EIQ values when incorporating uncertainty. Therefore, for example, a decision maker choosing acetamiprid over cypermethrin because of the nearly 8-unit difference in EIQs is choosing between two insecticides in which there may be no difference in EIQs when considering uncertainty (i.e., the EIQs overlapped 90% of the time in the simulation).

Our results demonstrate the problems with qualitative risk ratings in which uncertainty is not taken into account. Uncertainty cannot be ignored because the rating scores are surrogates for probabilities of occurrence or impact. However, the EIQ equation cannot take into account uncertainty the way that it is structured and provide reliable quotients of pesticide impact. As demonstrated by Cox, Babayev & Huber (2005) in general, and by us in particular, the EIQ equation contains layers of qualitative coding which results in loss of information and inconsistency in the interpretation of EIQ values.

In addition to the analyses above and those of Dushoff, Caldwell & Mohler (1994), the EIQ method is limited because it does not properly incorporate exposure. Therefore, the EIQ is inconsistent with the accepted notion of risk as a joint probability of toxicity and exposure. Because of this, the method essentially is a hazard rating scheme, not a risk rating scheme. The method roughly incorporates exposure by factoring scores for plant surface half-life, soil residue half-life, leaching potential, and surface runoff potential into the equation, but these factors that certainly influence exposure are proxies for exposure, not estimates of exposure. Similarly, the EIQ value is adjusted to a field-use EIQ by incorporating application rate of the pesticide and percent active ingredient in the formulation. This is particularly problematic because the adjustment to the EIQ based on application rate has nothing to do with resulting risk, only the amount of environmental loading of the pesticide. That is, a pesticide that is highly toxic at very low doses can have a low use rate with a concomitant low field-use EIQ even though the exposure is sufficient to cause unacceptable risks.

Cox, Babayev & Huber (2005), our findings presented here, and the conceptual problems pointed out by Dushoff, Caldwell & Mohler (1994), preclude the use of the EIQ or other pesticide risk ratings that are structured similarly to the EIQ. Dushoff, Caldwell & Mohler (1994) suggest various fixes, but many of these suggestions commit the same mathematical errors as the original EIQ scheme. In addition, different qualitative risk ranking systems can lead to different rankings of chemicals, and the discrepancy in rankings cannot be resolved unless different qualitative risk ranking systems are used together and evaluated, or a quantitative risk assessment is performed (Cox, Babayev & Huber, 2005; Morgan et al., 2000). The EIQ approach might have some utility for pesticides that are widely separated in EIQ values, such as diazinon versus carbaryl, but the conceptual problems with the scheme remain.

If the EIQ method and others like it are not conceptually or mathematically sound, then what should be used in their place? Risk is the joint probability of effect and exposure. In the case of pesticides, risk is the joint probability of toxicity and exposure. Therefore, for risk rating systems to be informative, toxicity and exposure must be integrated in an estimate of risk.

Risk rating systems for pesticides initially emerged when methods and models for estimating environmental exposure were in nascent stages of development. However, the ability to estimate the joint probability of exposure and toxicity (i.e., risk) currently is relatively simple and there are several acceptable models for estimating environmental exposures, e.g., FOCUS, PRZM-EXAMS, T-REX (FOCUS, 2001; USEPA, 2005a; USEPA, 2005b; USEPA, 2005c; USEPA, 2012).

The purpose of this article is not to examine a specific alternative to qualitative rating systems for pesticides. However, a starting point to create a useful quantitative rating system is the risk quotient (RQ) that is used in concept, but not necessarily by that specific term, by regulatory agencies throughout the world. An RQ is simply the ratio of estimated or actual environmental or dietary concentration of the pesticide to a toxic effect level or threshold. Some other terms for this ratio include hazard quotient (HQ), hazard index (HI), margin of safety (MOS), toxicity-exposure ratio (TER), and margin of exposure (MOE).

Peterson (2006) showed that an RQ approach is valuable for making direct comparisons of quantitative risks between pesticides. Furthermore, Peterson (2006) demonstrated that a numerical ranking of RQs for the purpose of comparing risks is valid across different levels of exposure refinement. Therefore, comparisons are equally valid whether using highly conservative exposure estimates (i.e., tier 1) or actual environmental exposures (tier 4). However, higher tiers should be used if the purpose is to accurately estimate the quantitative risk for an individual pesticide within a specific use and location scenario.

A risk rating system for pesticides is attractive and has potential benefits. However, our results suggest that qualitative rating systems should not be used for pesticide risk assessment, management, or decision making because they cannot properly discriminate between different levels of risk the way they are currently structured. We suggest that quantitative risk models be used for both risk assessment and risk management of pesticides.

Supplemental Information

Supplemental Information Spreadsheet data used in the simulation

Click here for additional data file.

We thank LG Higley and SH Hutchins for their reviews of earlier versions of this paper.

Additional Information and Declarations

Competing Interests

Author Contributions

The authors declare they have no competing interests.

Robert K.D. Peterson conceived and designed the experiments, performed the experiments, analyzed the data, contributed reagents/materials/analysis tools, wrote the paper, prepared figures and/or tables, reviewed drafts of the paper.

Jerome J. Schleier III analyzed the data, wrote the paper, reviewed drafts of the paper.

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
