# Peer review of "A probabilistic analysis reveals fundamental limitations with the environmental impact quotient and similar systems for rating pesticide risks"

_PeerJ, doi:10.7717/peerj.364_

## Round 0.1 · original submission · Major Revisions

Please find attached the comments of both reviewers, coming to similar conclusions. In particular, please ensure you react on the criticisms by both reviewers that the number and selection of compounds (nine insecticides) might be insufficient and/or might lead to a bias.

Reviewer 1 ·

Basic reporting

This manuscript attempts to demonstrate that environmental risk indicators as the Environmental Impact Quotient (EIQ), which are based on discrete and qualitative rating systems, reveal problems in rating the environmental risk. The authors demonstrate this by comparing results of the discrete risk rating based on the EIQ-Methodology with an approach using a probabilistic Monte Carlo simulations to calculate adjusted EIQ’s. They conclude that a variation in environmental effects, shown by discrete EIQ ratings may not be different when incorporating uncertainty.

The article is written in a very comprehensive language and describes the essential methods and results in a condensed way. The discussion and most of the conclusions are logical. Some of the conclusions need in my opinion to be undermined by more experimental and statistical analysis.

Experimental design

The submitted article is within the scope of the journal. In general the applied methods and models are described clearly and in a comprehensive manner so they can be reproduced by other scientists.
The method of mapping discrete risk ratings to probabilities for four factors within the EIQ method seems logic and is certainly a new approach.
Despite this, the authors do not justify their choice of active ingredients used in the analysis to compare the adjusted and unadjusted EIQ’s. The authors should explain why were only nine insecticides were chosen from the possible list of 158 insecticides or 480 active ingredients in the cited in the database? Why was the range for the selected ai’s limited to EIQ’s of 13.3 to 44.4 (total range 8,67 to 101,83) to prove the hypothesis of the article.

Line 102 : Maybe mention here also what constant values were used for the score of 1 and 5.

Validity of the findings

The criticism that through qualitative categorization of quantitative information like toxicity and other active ingredient properties will results in a loss of information and might lead to wrong ratings of risk is appropriate. But in the article this is demonstrated only for a set of nine insecticides.

Overall the statistical analysis of the comparison of the two approaches is weak. The conclusion that a variation in environmental effects, shown by discrete EIQ ratings may not be different when incorporating uncertainty is demonstrated for only one pair of insecticide combinations. The difference of the risk ratings of the probabilistic EIQ approach with 10% is clearly less than the 21% difference of the original EIQ method. Therefore the adjusted EIQ approach suggests that there is a reduced environmental impact. But I can’t follow that the reduced difference leads to the conclusion that there may be no difference (line 117).

The authors should also consider to show the comparisons of other insecticide pairs (e.g. in a Table). It should be mentioned that the difference between the two approaches is less for most of the other insecticide pairs and that for some insecticide pairs the adjusted EIQ shows larger differences (+7% for dimethoate and dimethoate) than the original EIQ method.

The conclusion that the EIQ method is a problematic method to assess the risk of pesticides since it can not consider uncertainty should be clearly linked to the results.

Since risk assessment methods based on discrete categorization of a.i. properties, like the EIQ, are often used to rank pesticides the authors may also consider to include a rank analysis of the two methods.

Line 111-112
This is in my eyes this not the correct interpretation of the results. There is no overlap (10th percentile of ai1 > 90th percentile of ai2) between the EIQ’s of lambda-cyhalothrin/bifemithrin/ cyfluthrin and permethrin/malathion/acetamiprid. The authors should clearly define what they mean by overlap and which insecticides overlap.

Line112-113
The authors should mention that the difference between the two approaches is less for most of the other insecticide pairs and that for some insecticide pairs the adjusted EIQ shows larger differences (+7% for dimethoate and dimethoate) than the original EIQ method.

Figure 1
The values of the unadjusted EIQ for malathion, permethrin and bifenthrin do not correspond to the values in excel sheet of the supplementary material.

Reviewer 2 ·

Basic reporting

I believe that the authors have given more credence to the EIQ process as a risk assessment tool. It was not intended for this purpose and is not used for this purpose by regulatory agencies. It is similar in many ways to the hazard ranks associated with toxicity categories and pictograms and provides a useful tool in decision making by managers of pests. In this the paper is somewhat unfair in its criticism and they are certainly not the first to point this out. If this criticism was removed or toned down and the incorporation of risk treated as a refinement, the paper would be acceptable. However, before accepting it, more exploration of other scales for the PEIQ should be explored and the reasons for the deviation in Fig 1 explained.

Experimental design

The authors should conduct a correlation between their PEIQ and the EIQ to test extrapolation and they should include more than 9 chemicals in their assessment. There are many listed in the EIQ website to choose from

Validity of the findings

The findings are what would be expected and have been noted by others already.

Additional comments

There are a number of specific comments in the attached PDF of the paper. These need to be addressed.

Annotated reviews are not available for download in order to protect the identity of reviewers who chose to remain anonymous.

---

## Round 0.2 · accepted · Accept

Thanks for the extensive and convincing rebuttal! I guess not everybody will agree with the paper in the end. Which perhaps is actually a good thing, if the paper manages to spark a good debate...